# Characteristics Prediction and Optimization of GaN CAVET Using a Novel Physics-Guided Machine Learning Method

**DOI:** 10.3390/mi16091005

**Published:** 2025-08-30

**Authors:** Wenbo Wu, Jie Wang, Jiangtao Su, Zhanfei Chen, Zhiping Yu

**Affiliations:** 1Innovation Center for Electronic Design Automation Technology, Hangzhou Dianzi University, Hangzhou 310018, China; 251040086@hdu.edu.cn (W.W.); jtsu@hdu.edu.cn (J.S.); 2Ningbo Institute of Digital Twin, Eastern Institute of Technology, Ningbo 315200, China; zchen@idt.eitech.edu.cn; 3School of Integrated Circuits, Tsinghua University, Beijing 100084, China; yuzhip@tsinghua.edu.cn

**Keywords:** current aperture vertical electron transistor, TCAD, physics-guided machine learning, shallow neural network, hypernetwork, shortcut, physics-guided artificial neural network

## Abstract

This paper presents a physics-guided machine learning (PGML) approach to model the I–V characteristics of GaN current aperture vertical field effect transistors (CAVET). By adopting the method of transfer learning and the shortcut structure, a physically guided neural network model is established. The shallow neural network with tanh as the basis function is combined with a hypernetwork that dynamically generates its weight parameters. The influence of transconductance is added to the loss function. This model can synchronously predict the output and transfer characteristics of the device. Under the condition of small samples, the prediction error is controlled within 5%, and the $R^{2}$ value reaches above 0.99. The proposed PGML approach outperforms conventional approaches, ensuring physically meaningful and robust predictions for device optimization and circuit-level simulations.

## 1. Introduction

Gallium Nitride (GaN) [1,2] is increasingly becoming a research hotspot in the field of power electronics due to its excellent properties, such as high electron mobility, high electron saturation velocity, high-temperature resistance, and high thermal conductivity. Vertical GaN power transistors are emerging as a key solution for next-generation power converters. Among them, the Current Aperture Vertical Electron Transistor (CAVET) [3,4] is an effective combination of lateral and vertical topologies, which can take advantage of the two-dimensional electron gas (2DEG) from high-electron-mobility transistors (HEMT) with a vertical device structure design, enabling higher breakdown voltage without increasing the chip area. Meanwhile, CAVETs do not suffer from surface state-related dispersion, and they perform better thermally than HEMTs due to homo-epitaxially grown layers. But due to the complex geometry of CAVET, the design rules are non-trivial, and the modeling is inherently complicated.

In the development of CAVET devices and their models, it is essential not only to refine device fabrication processes [1,2] to enhance performance, but also to conduct extensive simulation-based research for structural optimization, ensuring suitability for high-frequency, high-power, and other application scenarios. The optimized designs for the CAVET use two-dimensional (2-D) technology computer-aided design (TCAD) simulations [5]. Recently, machine learning has developed rapidly due to its capability of detecting patterns and making predictions, leading to the continuous proposal of many novel machine learning-based methods [6,7]. The methods that combine TCAD with machine learning (ML), known as TCAD-augmented ML (AML) [8,9,10,11,12,13,14,15,16,17], have been widely adopted for various applications in electronic design automation. TCAD-AML leverages TCAD data to train advanced models for defect analysis [8,9], predicting device characteristics [10,11], inverse design [12,13], developing surrogate models [14,15], exploring device manifold learning [16], and reconstructing electrical characteristics [17]. This approach is extensively employed to uncover hidden correlations between electronic parameters and electrical performance with high accuracy and efficiency. The potential of TCAD-AML holds significant promise for early-stage device design, and the lack of physical interpretability diminishes the effectiveness of TCAD-AML. Currently, lateral GaN HEMT devices have standard CMC models such as the ASM model [18] using a surface potential (SP)-based approach to capture terminal characteristics in GaN HEMTs by solving the Schroedinger–Poisson coupled equations. Next is the MIT Virtual Source GaN model [19] originally developed for highly scaled Si-FETs with a quasi-ballistic mode of transport, adopting a different interpretation of the carrier velocity by using an empirical saturation function for GaN HEMTs. Vertical MOSFET devices are characterized by standard CMC models like the PSP model [20] and the BSIM model [21]. In the PSP model [20], the diffusion and drift currents are expressed in terms of the surface potential. These equations are valid for all levels of inversion. The BSIM model [21] is a threshold-voltage-based model, which has non-intersecting expressions for the diffusion current in the subthreshold region and the drift current in the strong inversion region. The CAVET device, with its complex hybrid lateral and vertical structure, cannot be accurately characterized by these existing models [18,19,20,21].

To address the limitations of existing GaN CAVET (as shown in Figure 1) modeling technologies, we present the prediction of characteristics and parameter optimization of the device using the physics-guided machine learning (PGML) approach [22]. The PGML approach integrates the *Id*(*Vgs*, *Vds*) formula governed by the tanh function for nonlinear behavior [23,24,25,26,27] into the physics-guided artificial neural network (PG-ANN) architecture, ensuring outputs adhere to physical laws. The PG-ANN model combined a shallow neural network [28] with a hypernetwork [29] and introduced a shortcut [30], which was employed to train the model using data generated from SILVACO ATLAS simulations [31], effectively capturing the I–V characteristics of the devices. The shallow neural network leverages a linear combination of multiple tanh activation functions to achieve a nonlinear mapping from input to output. Instead of directly learning the weight parameters, the hypernetwork learns a function to dynamically generate these parameters, enhancing the model’s generalization ability. A physics-guided multi-objective loss function is designed to avoid overfitting by incorporating error terms for transconductance curves. This loss function ensures the accurate prediction of I–V characteristic curves, making the neural network model suitable for small datasets. During the learning process of the PG-ANN model, the weight parameters dynamically generated by the hypernetwork effectively utilize the hidden information within the dataset, improving the model’s learning efficiency. These weight parameters serve as the linear combination parameters for the tanh activation function, enabling the model to quickly capture the relationship between device input and output characteristics, which is particularly effective for analog circuit simulations.

This paper is arranged as follows: In Section 2, we present the TCAD simulation and sample generation. Section 3 details the PGML approach and PG-ANN model, and TCAD validation is shown in Section 4. Finally, we conclude this paper in Section 5.

## 2. TCAD Simulations of GaN CAVET

TCAD computations of $I_{ds}-V_{ds}$ and $I_{ds}-V_{gs}$ characteristics of CAVET are used to train the ANN model. We employed experimentally calibrated SILVACO example ganfetex20, which is based on the reference paper [32], to generate I–V characteristics by adjusting the following key parameters: (1) aperture layer length $L_{ap}$; (2) gate overlap length $L_{go}$; (3) unintentionally doped GaN layer thickness $T_{uid}$; (4) current blocking layer thickness $T_{cbl}$; (5) *p*-type doping of the current blocking layer $N_{cbl}$; (6) drift layer thickness $T_{drift}$; and (7) doping of the drift layer $N_{drift}$.

This paper employed TCAD simulations of 3^7^ = 2187 samples, varying $L_{ap}$, $L_{go}$, $T_{uid}$, $T_{cbl}$, $N_{cbl}$, $T_{drift}$, and $N_{drift}$. Each device included the setting parameters gate voltage $V_{gs}$, output drain voltage $V_{ds}$ as inputs for the ANN, with the drain current $I_{ds}$ serving as the output. The specific TCAD settings are outlined in Table 1. For each sample, representing a distinct device with unique geometry and effective doping concentration, TCAD simulations were conducted to obtain its I–V characteristics. We systematically changed the inputs and obtained the corresponding outputs for each sample. The ANN models were then trained using these samples.

## 3. The PGML Approach and PG-ANN Model

In network model building, given the small data volume and low parameter dimension, the network structure should be concise to fully mine sample information. An innovative PG-ANN framework was established to simultaneously predict transfer and output characteristics. For the PG-ANN-based prediction of I–V characteristics, the network inputs are $V_{gs}$, output $V_{ds}$, and transistor parameters (see Table 1), with $I_{ds}$ as the output. The process of implementing PGML involves the following components: (1) designing the PG-ANN model; (2) defining an appropriate loss function of PG-ANN models; (3) training the PG-ANN model; and (4) screening of the PG-ANN model will be discussed in this section.

### 3.1. Designing the PG-ANN Model

PG-ANN modeling approach uses a combination of a simple neural-network structure, such as the shallow neural network model in Figure 2, and the hypernetwork in Figure 3. Shortcuts are also introduced.

The constructed PG-ANN model mainly uses the tanh function as the activation function, which ensures physical consistency. Its parameters are dynamically generated by a hypernetwork. Cross multiplication and division layers enhance the model’s expressiveness while cutting parameters by an order of magnitude. A new PG-ANN model demonstrates enhanced capability over our prior physics-inspired ANN [33], enabling concurrent prediction of output and transfer characteristics through integrated transfer learning and shortcut connections.

#### 3.1.1. Designing the Shallow Neural Network Model

The Curtice model [25] is a nonlinear empirical model for describing the behavior of field-effect transistors (FETs). It employs a hyperbolic tangent tanh function to characterize the variation in drain current $I_{ds}$ with drain voltage $V_{ds}$, enabling accurate representation of the FET’s transconductance nonlinearity.(1)$$I_{ds}=I_{ds0}\left(1+\lambda V_{ds}\right)\tanh\left(\alpha V_{ds}\right)$$

The nonlinear approximation capability of the tanh function effectively characterizes device I–V behavior, thereby significantly improving the model’s representation of key device physics effects. Accordingly, the novel PG-ANN model employs a shallow neural network architecture that implements $V_{ds}$ to $I_{ds}$ mapping through a linear combination of multiple tanh activation functions.(2)$$I_{ds}=a_{1}\tanh\left(a_{2}V_{ds}+a_{3}\right)+a_{4}\tanh\left(a_{5}V_{ds}+a_{6}\right)+a_{7}\tanh\left(a_{8}V_{ds}+a_{9}\right)+a_{10}$$

The compact shallow neural network structures designed based on Equation (3) offer higher computational efficiency and strong physical interpretability. This helps understand the relationship between model parameters and device electrical behavior.

#### 3.1.2. Designing the Hypernetwork Model

We use a hypernetwork model to dynamically generate weights $a_{n}$ (n = 1~10), which is the weight and bias in the PG-ANN for the shallow neural network. Hypernetwork is a lightweight hypernetwork that dynamically generates the weight parameters of the target network in real-time based on input key device structural parameters, as illustrated in Figure 3, rather than assigning fixed weights. This approach enhances the model’s fitting capability while typically requiring fewer parameters, making it more suitable for training with small-sample data.

Given the proportional relationship between voltage and fitting parameters, we incorporate cross-division structures. This creates an effective network structure where each parameter, influenced similarly by voltage changes when others are constant, is represented by a new neural network with strong functional expressiveness. This network shows strong adaptability to the task and yields good results. Also, dropout layers are added to boost model robustness and prevent overfitting, as shown in Figure 3.

#### 3.1.3. Integration of Shallow Neural Network and Hypernetwork

Initially, a shallow neural network performs curve fitting on raw data to obtain initial weights $a_{n}$ (n = 1~10), which are used to pre-train a hypernetwork. The hypernetwork dynamically generates weights for the PG-ANN using key device structural parameters like $L_{ap}$, $L_{go}$, $T_{uid}$, $T_{cbl}$, ${\;N}_{cbl}$, $T_{drift}$, and $N_{drift}$. This establishes a physical constraint mapping between voltage and current.

In this architecture, the PG-ANN leverages hypernetwork-generated parametric weights and real-time drain current shape features. It utilizes the neural network’s fitting capability to create a device I–V characteristic prediction model. Pre-training the hypernetwork prevents local optima, speeds up fitting, and enhances fitting efficiency. A hybrid optimization approach combines genetic algorithms for initial parameter searching to locate different extreme points and gradient descent algorithms to quickly find local optima. This accelerates training significantly.

#### 3.1.4. Extension of PG-ANN

Designing a new drain current $I_{ds}$, as in Equation (3), can simultaneously output the transfer and output characteristics of CAVET. As shown in Equation (3), which incorporates the $V_{gs}$, enabling comprehensive I–V modeling within a unified neural network architecture.(3)$$I_{ds}=relu{[c}_{1}tanh\left(k_{1}*V_{ds}\right)tanh\left(a_{1}V_{gs}+a_{2}\right)+c_{2}tanh\left(k_{2}V_{ds}\right)tanh\left(a_{3}V_{gs}+a_{4}\right)\;+c_{3}tanh\left(V_{ds}\right)V_{gs}softplus\left(k_{3}V_{gs}{para}_{10}+k_{4}\right)]$$

In this equation, as the two networks share the same hypernet, the ${para}_{10}\;$ is the original position of the $PG-ANN^{\prime }s$ $a_{10}$. $k_{i}$ is independent of $V_{ds}$, and $a_{i}$ is independent of $V_{gs}$, the ${para}_{10}$ is the product of $a_{8},a_{9}\;in\;Equation\;(2)$.

Some parameter combinations exhibit significant nonideal effects and are nearly independent of $V_{gs}$. To address this, we introduce a shortcut, inspired by ResNet. By summing the inputs and outputs of a shallow neural network, these connections help preserve parameter characteristics in deep structures and reduce gradient vanishing. Experiments show that shortcuts enhance the network’s learning ability. Additionally, a small, separate network can fit these non-ideal effects. Since large non-ideal effects are undesirable, the outputs of the shortcut can assess parameter–combination quality.

### 3.2. The Loss Function of PG-ANN Models

For the IV characteristics, device transconductance from TCAD simulations is introduced as prior physical knowledge in dataset building to aid model training. This enhances the model’s understanding of device physics and reduces overfitting, improving generalization and prediction accuracy. The loss function includes the mean squared error (4)$$MSE\left(I_{d,TCAD},I_{d,pred}\right)=\frac{{\left(I_{ds}-I_{ds,pred}\right)}^{2}}{N}$$

To guide the learning of data science models to physically consistent solutions, we introduce a physics-based loss function in Equation (5) which aims to introduce physical constraints so that the model can learn data characteristics while also following physical laws. Let us denote the physical relationships between the target variable $I_{d}$ and other variables $V_{gs}$ and $V_{ds}$ using their first order derivative $g_{x}=\partial I_{d}/V_{x}$ the gate transconductance ($g_{m}=\partial I_{d}/V_{gs}$, the output conductance $g_{d}=\partial I_{d}/V_{ds}$).(5)$$MSE\left(g_{x,TCAD},g_{m,pred}\right)=\frac{{\left(g_{x}-g_{x,pred}\right)}^{2}\;}{N}$$

These physics-based derivatives must meet the same criteria as other loss function terms (i.e., continuous and differentiable). One way to measure if these physics-based derivatives are being violated in the PG-ANN model predictions is to evaluate the following physics-based loss function $Loss\left(I_{ds},I_{ds,pred}\right)$(6)$$Loss\left(I_{ds},I_{ds,pred}\right)=\left\{\begin{array}{c} 0*\left[{\left(I_{ds}-I_{ds,pred}\right)}^{2}+{k\left(g_{x}-g_{x,pred}\right)}^{2}\right]\;,\left|I_{ds}-I_{ds,pred}\right|<10^{-8}\\ 1*\left[{\left(I_{ds}-I_{ds,pred}\right)}^{2}+{k\left(g_{x}-g_{x,pred}\right)}^{2}\right]\;\;,\left|I_{ds}-I_{ds,pred}\right|\geq 10^{-8} \end{array}\right.$$(7)$${Loss}_{MSE}=\frac{\sum Loss\left(I_{ds},I_{ds,pred}\right)}{n}$$

Since known physical laws are considered applicable to any unseen data instances, the PG-ANN model adopts the physical consistency of the first-order partial derivative of device current with respect to voltage as a learning objective. Even with limited and unrepresentative training data, this approach still achieves better generalization capability. Furthermore, by incorporating this derivative into the device design workflow, the PG-ANN model accelerates the device optimization process.

### 3.3. PG-ANN Model Training

#### 3.3.1. Pre-Training of Shallow Neural Network

Fit the output characteristic curves under various parameters. For instance, with 7 variables each having 3 possibilities, 3^7^ shallow neural networks need pre-training. To speed this up, determine initial parameter solutions using specific methods. Given the large parameter variations, design the learning rate carefully. Start with a small initial rate and retain pre-feedback parameters. Combine gradient descent and genetic algorithms by first optimizing with the genetic algorithm, then applying gradient descent. Check the fitting quality, which is the product of output characteristic fitting and transconductance fitting. If unsatisfactory, alternate between the genetic and gradient descent algorithms. If results are still poor after multiple tries, output and flag them directly. During training, use PyTorch2.5.1’s grad attribute to obtain the model’s predicted transconductance, compare it with TCAD-simulated values, and calculate the loss function. To boost robustness to data errors, introduce an error tolerance mechanism: if the prediction error is less than 10^−8^, do not increase the loss function. This prevents tiny actual measurement errors from disrupting training.

#### 3.3.2. Pre-Training of Hypernetwork

The computational load of this phase is relatively low, as its primary purpose is to obtain a robust initial model that serves as the foundation for subsequent full training. We employ a simple yet effective loss function—the sum of squared differences in the PG-ANN network parameters—which is used as the loss for the hypernetwork. Although the hypernetwork architecture includes complex operations, its gradient calculation remains concise. Since only a single network is optimized at this stage, gradient descent can efficiently achieve a well-converged initial solution. The hypernetwork incorporates layers involving multiplication and division, inspired by symbolic regression; however, the selection of these operations is manually determined to balance expressiveness and stability.

Moreover, to enhance the model’s generalization ability during initial training, we have introduced a dropout layer. Its random deactivation mechanism effectively alleviates overfitting and improves the network’s robustness under different input conditions.

#### 3.3.3. Combined Training of Shallow Neural Network and Hypernetwork

This study integrates a shallow neural network with a hypernetwork for device modeling. The structural parameters of the device are first fed into the hypernetwork, which generates the weight factors for the shallow neural network. The latter then takes the voltage parameters as inputs to predict the drain current. The training process is divided into two stages: in the first stage, the hypernetwork is trained using parameters obtained from the PG-ANN pre-training, after which its parameters are frozen and the expansion layer is trained to initialize its values. In the second stage, the hypernetwork is unfrozen, and the entire network undergoes full training. The same loss function as used in the pre-training phase is applied, enhancing the PG-ANN’s generalization and robustness against interference in device modeling.

The parameter space becomes complex after combining the genetic algorithm hyperparameters for optimizing the shallow neural network alone differ greatly, so the optimization strategy must be readjusted based on the model structure. Moreover, the retained multiple generations’ historical optimal solutions effectively prevent search deviation from the optimal region, which can improve overall search stability and convergence efficiency.

### 3.4. Model Screening

The dataset is divided into training and test sets with a sample ratio of 7:3. For parameter optimization, the Adam optimizer is selected due to the significant variation in parameters. A smaller initial learning rate is chosen, and parameters before feedback are retained. If the loss function increases after feedback, the learning rate is reduced threefold, and the saved parameters are overwritten. If it decreases, the learning rate increases by 1% to prevent it from becoming too small. The batch size is set to 128. Testing shows that the optimal network performance is achieved at 5000 training steps.

## 4. Results and Discussion

For the PG-ANN model validation, key aspects like anti-jamming capability, computational efficiency, and fitting accuracy are assessed.

### 4.1. Anti-Jamming Capability of the PG-ANN Model

The PG-ANN’s structure is concise, so it will not fluctuate much and can ignore unstable variations from TCAD simulations. This prevents overfitting and insulates the model from interference caused by abnormalities, as shown in Figure 4. 

### 4.2. PG-ANN Model Accuracy

In Figure 5, the scatter plot of all data points shows that the scatters fit the plots just fine, with all points closely clustered around the ideal line x = y. This indicates that the PG-ANN model has high predictive accuracy.

Representative samples [32] of output and transfer characteristic curves were chosen in Figure 6. The curves formed by true values and the scatter points of predicted values are plotted together. This clearly shows the model’s fitting performance under different characteristics.

## 5. Conclusions

In this work, we proposed a PGML approach for accurately predicting the electrical characteristics of GaN CAVET. By leveraging TCAD simulations, we generated a small dataset encompassing a wide range of device parameters and utilized this dataset to train a PG-ANN model with embedded physical constraints.

For the device I–V characteristic prediction, a PG-ANN model combining a Shallow neural network, a hypernetwork, and shortcut connections is proposed. A physics-based loss function is introduced to guide the learning process of the PG-ANN model. This loss function takes transconductance into account, enabling better generalization when training data is limited and not fully representative. It effectively prevents overfitting in neural networks and accelerates the convergence of functions.

Results demonstrated that our model achieves high precision in predicting I–V characteristics, effectively capturing the influence of device parameters on electrical performance. Even with limited samples, the model keeps prediction errors within 3.3% and achieves an $R^{2}$ of over 0.999951. The lightweight architecture of our model also enables efficient training while maintaining accuracy, making it particularly well-suited for few-shot learning scenarios and the prediction of device characteristics.

## Figures and Tables

**Figure 1 micromachines-16-01005-f001:**
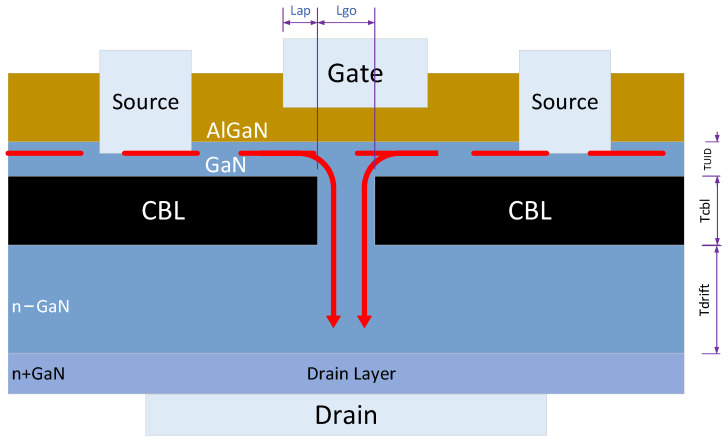
Schematic showing the epi-structure of the CAVET consisting of the AIGaN/GaN heterojunction channel, the *p*-GaN current blocking layer (CBL) and aperture layer, and n-GaN drift region.

**Figure 2 micromachines-16-01005-f002:**
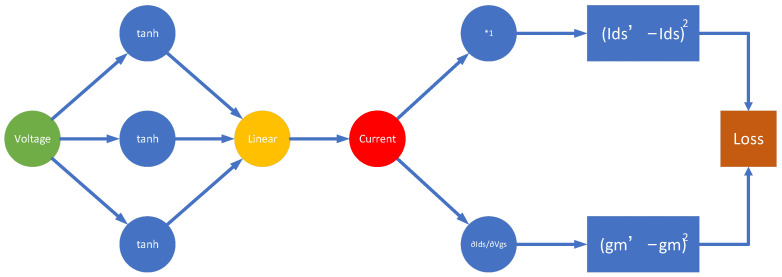
Diagram of shallow neural network.

**Figure 3 micromachines-16-01005-f003:**
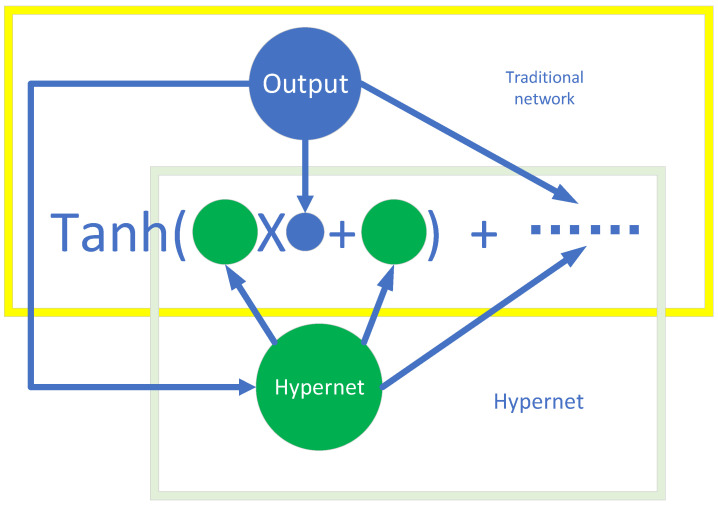
Diagram of hypernetwork.

**Figure 4 micromachines-16-01005-f004:**
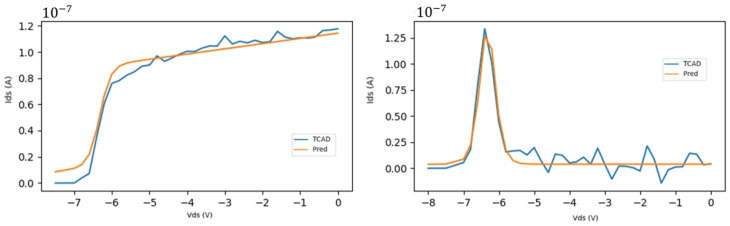
Fluctuations caused by TCAD simulation errors.

**Figure 5 micromachines-16-01005-f005:**
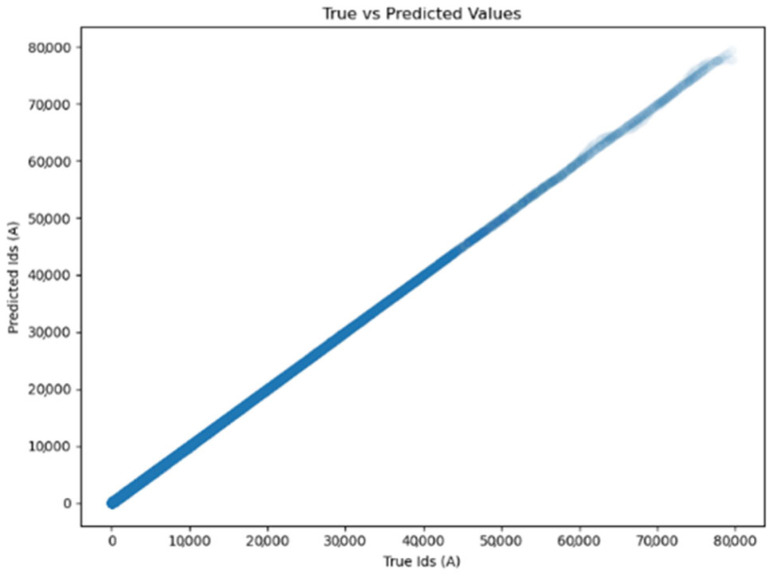
Correlation scatter plot.

**Figure 6 micromachines-16-01005-f006:**
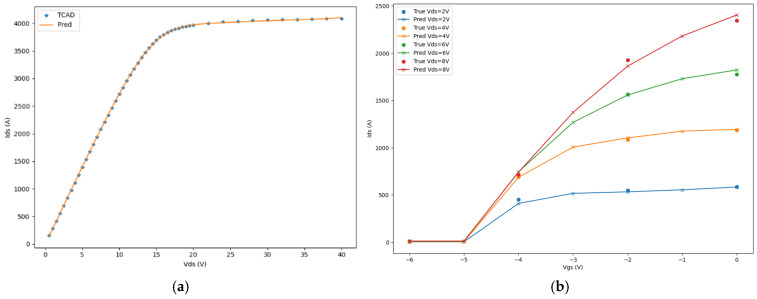
Calculated and measured I–V characteristics for a CAVFT. (**a**) Output characteristic $V_{gs}=0\;V.\;$(**b**) Transfer characteristic $V_{ds}=\;$2 V to 8 V with a 2 V step. (Solid line.) The results of the presented SP-based model are compared with (scatter point) the Vth-based model and (symbol) the experimental data [32].

**Table 1 micromachines-16-01005-t001:** TCAD parameter setting.

Parameter	Range (Min, Max)	Step
$L_{ap\;}$ [μm]	(1, 4, 10, 15)	
$L_{go}\;[\mu m]$	(1, 3, 5)	2
$T_{cbl}\;[\mu m]$	(0.1, 0.3, 0.5)	0.2
$N_{cbl}\;$ [cm^−3^]	(8 × 10^16^, 2 × 10^17^, 8 × 10^17^)	
$N_{drift}\;$ [cm^−3^]	(2 × 10^15^, 2 × 10^16^, 2 × 10^17^)	
$T_{uid}\;$ [μm]	(0.1, 0.15, 0.2)	0.05
$T_{drift}$ [μm]	(1, 3, 5)	2
$V_{gs\;}$ [V]	(−8, 0)	0.2
$V_{ds}\;$ [V]	(0, 40)	0.5

## Data Availability

The data presented in this study were generated using TCAD simulations and are not publicly available. However, the typical simulation setup and parameters are described in the article, and similar data can be reproduced by running simulations under equivalent conditions. Further details or raw data may be available from the corresponding author upon reasonable request.

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
