# Peer review of "Characteristics Prediction and Optimization of GaN CAVET Using a Novel Physics-Guided Machine Learning Method"

_micromachines, 2025, doi:10.3390/mi16091005_

Round 1

Reviewer 1 Report

Comments and Suggestions for Authors
  1. The inclusion of transconductance (g) in the loss function is interesting. However, the authors should provide more detailed physical justification for why g is chosen over other derivatives (e.g., output conductance) and how this specifically enforces physical consistency in the context of GaN CAVET behavior.
  2. The hypernetwork is a key component of the model, but its architecture (number of layers, neurons, activation functions) is not described. Please provide more details on its design and how it interacts with the shallow network during training, especially regarding parameter sharing and gradient flow.
  3. Vertical field effect transistorsare not the commonly used device structure for GaN. The AlGaN/GaN heterojunction (Applied Surface Science 2017, 423, 675-679) and HEMT (Materials & Design, 2018, 148, 1-7) are the frequently reported techniques. The authors should make a comment on the reported techniques at least in the introduction portion.

Author Response

Thank you for your valuable time and insightful comments on our manuscript. We have carefully considered all the suggestions and have revised the manuscript accordingly. A detailed point-by-point response to all comments has been provided in the attached file, "reviewers1.pdf".

We believe that the manuscript has been significantly improved by these revisions. We have addressed all the reviewers' concerns, and we hope the revised manuscript is now suitable for publication in your journal.

Reviewer 2 Report

Comments and Suggestions for Authors

This study develops a physics-guided machine learning (PGML) framework that integrates shallow neural networks, hypernetworks, and physics-informed loss functions to accurately predict the I–V characteristics of GaN CAVET devices. By embedding transconductance constraints and leveraging small-sample TCAD data, the proposed model achieves high predictive accuracy (>0.99 R²) and robustness, making it suitable for device optimization and circuit-level simulations.

Here are my comments:

Results are only compared with TCAD-generated data. No experimental validation is provided.

The extremely high reported R² (>0.9999) likely reflects overfitting to the synthetic TCAD dataset rather than true predictive capability.

The dataset size (2187 samples) is relatively small but highly structured. The study does not investigate how the model scales with noisy or non-uniform data.

The manuscript claims lightweight training, but provides no time complexity analysis or hardware cost comparison against standard ANN or CNN approaches.

Several important cross-disciplinary literatures are missing, like Predicting flow status of a flexible rectifier using cognitive computing; and Bio-inspired circular soft actuators for simulating defecation process of human rectum.

Although transconductance is incorporated, higher-order physical constraints (e.g., capacitance, breakdown behavior, thermal effects) are absent, leaving the model incomplete for device-level physics.

Formatting issues (e.g., Equation (3), unclear parameters such as “para10”) reduce clarity and reproducibility.

No error tolerance discussion beyond 10⁻⁸. This arbitrary threshold is unrealistic in experimental scenarios where noise dominates.

Citations [23] SILVACO link are not peer-reviewed, reducing academic rigor.

The claim of “real-time circuit simulation readiness” is unsupported by any demonstration.

Author Response

Thank you for your valuable time and insightful comments on our manuscript. We have carefully considered all the suggestions and have revised the manuscript accordingly. A detailed point-by-point response to all comments has been provided in the attached file, "reviewers2.pdf".

We believe that the manuscript has been significantly improved by these revisions. We have addressed all the reviewers' concerns, and we hope the revised manuscript is now suitable for publication in your journal.

Round 2

Reviewer 1 Report

Comments and Suggestions for Authors

I suggest to accept this manuscript.

Reviewer 2 Report

Comments and Suggestions for Authors

I have reviewed the revised manuscript and I am pleased to report that the authors have made substantial improvements in response to my previous comments. The manuscript now addresses the key concerns I raised, and the revisions enhance both the clarity and depth of the work. I recommend that the manuscript be accepted for publication in its current form.